# Knowledge Distillation $\approx$ Label Smoothing: Fact or Fallacy?

**Md Arafat Sultan**
IBM Research AI
arafat.sultan@ibm.com

## Abstract

Originally proposed as a method for knowledge transfer from one model to another, some recent studies have suggested that knowledge distillation (KD) is in fact a form of regularization. Perhaps the strongest argument of all for this new perspective comes from its apparent similarities with label smoothing (LS). Here we re-examine this stated equivalence between the two methods by comparing the predictive confidences of the models they train. Experiments on four text classification tasks involving models of different sizes show that: (a) In most settings, KD and LS drive model confidence in completely opposite directions, and (b) In KD, the student inherits not only its knowledge but also its confidence from the teacher, reinforcing the classical knowledge transfer view.

## 1 Introduction

Knowledge distillation (KD) was originally proposed as a mechanism for a small and lightweight *student* model to learn to perform a task, *e.g.*, classification, from a higher-capacity *teacher* model (Hinton et al., 2014). In recent years, however, this view of KD as a knowledge transfer process has come into question, with a number of studies finding that it has features characteristic of a regularizer (Yuan et al., 2020; Zhang and R. Sabuncu, 2020; Dong et al., 2019). Perhaps the most direct argument for this new position comes from Yuan et al. (2020), who show that KD has interesting similarities with label smoothing (LS) (Szegedy et al., 2016). One of their main conclusions, that "KD is a type of learned LS regularization," has subsequently been reiterated by many (Menon et al., 2021; Chen et al., 2021a; Shen et al., 2021; Chen et al., 2021b) and remains an important part of our understanding of KD to this day.

This is a rather curious suggestion, however, given the starkly different purposes for which the two methods were originally developed: LS was designed *"for encouraging the model to be less confident"* (Szegedy et al., 2016) by imposing a uniform prior on the target distribution; the resulting new distribution is intended to simply prevent overfitting without revealing any new details about the task. KD, on the other hand, was designed to explicitly teach the student better inter-class relationships for classification, so that *"a BMW, for example, may only have a very small chance of being mistaken for a garbage truck, but that mistake is still many times more probable than mistaking it for a carrot"* (Hinton et al., 2014).

Such different motivations of the two methods prompt us to re-examine the claim of their equivalence in this paper, where we find the underlying argument to be essentially flawed. With a growing dependence in key areas like NLP on increasingly large models, KD and other model compression methods are becoming more relevant than ever before (Wang et al., 2023; Taori et al., 2023; Dettmers et al., 2023). It is therefore crucial that we also interpret such methods correctly to help steer their future research in the right direction, which is a primary goal of this work.

KD and LS do indeed share the common property that they both substitute one-hot training labels with *soft* target distributions. The key question for us, however, is whether the soft teacher estimates in KD simply regularize training like the smoothed labels in LS do, or if they present a detailed and finer-grained view of the task, as originally intended. Yuan et al. (2020) argue in favor of the former based on observations such as (a) similarities in the cost functions of KD and LS, and (b) strong generalization of distilled image classifiers independent of the level of expertise of their teachers.

Here we first analyze their arguments and point out why they may be inadequate for a "KD $\approx$ LS" assertion (§2, 3). Then we present empirical evidence on four text classification tasks from GLUE (Wang et al., 2019) that KD *lacks the defining prop-*

*erty of* LS in most settings. Concretely, relative to standard empirical risk minimization (ERM) over one-hot expert labels, LS *by design* increases the uncertainty (entropy) in the trained model's posterior over the associated categories. In other words, it always reduces the model's confidence in its predictions, as expected. The effect of KD on the student's posterior, on the other hand, is a function of the relative capacities of the teacher and the student: In the more common setting where a larger teacher is distilled into a smaller student and also in self-distillation, KD almost always reduces entropy over ERM, while also improving accuracy. It is only when the student is larger than its teacher that we see a small increase in entropy—still much less than in LS—along with a smaller gain in accuracy. These results clearly show that unlike LS, KD does not operate by merely making the trained model less confident in its predictions.

The following is a summary of our contributions:

- We study the inner workings of KD in the relatively under-explored modality of language, specifically on text classification tasks.

- We show that KD lacks the defining property of LS regularization, rebutting the claim that the former is a form of the latter.

- We demonstrate how the specific selection of the teacher model determines student confidence in KD, providing strong support for its classical knowledge transfer view.

## 2 The Argument for KD ≈ LS

Let us first take a look at the main arguments, primarily from Yuan et al. (2020), for why KD should be considered a form of LS. Given a training instance $x$, let $q(k|x)$ be the corresponding one-hot ground truth distribution over the different classes $k \in \{1, ..., K\}$. LS replaces $q(k|x)$ with the following smoothed mixture as the target for training:

$$q_{ls}(k|x) = (1 - \alpha)q(k|x) + \alpha u(k)$$

where $u(k) = 1/K$ is the uniform distribution over the $K$ classes. Yuan et al. (2020) show that optimizing for $q_{ls}$ is equivalent to minimizing the following cost function:

$$\mathcal{L}_{LS} = (1 - \alpha)H(q, p) + \alpha D_{KL}(u, p) \quad (1)$$

where $p(k|x)$ is the model output for $x$, and $H$ and $D_{KL}$ are the cross-entropy and the Kullback-Leibler divergence between the two argument distributions, respectively. Note that the $H(q, p)$ term is

the same cross-entropy loss $\mathcal{L}_{ERM}$ that is typically minimized in standard empirical risk minimization (ERM) training of classifiers.

KD, on the other hand, is defined directly as minimizing the following cost function:

$$\mathcal{L}_{KD} = (1 - \alpha)H(q, p) + \alpha D_{KL}(p^t, p) \quad (2)$$

where $p^t(k|x)$ is the posterior computed by a pre-trained teacher $t$ given $x$. The similarity in the forms of Eqs. 1 and 2 is one of the main reasons why Yuan et al. (2020) interpret KD "as a special case of LS regularization."

A second argument stems from their observation on a number of image classification datasets that a "teacher-free KD" method, which employs a smoothed target distribution as a virtual teacher, has similar performance as standard KD. Finally, some of their empirical observations, where KD improves student accuracy even when the teacher is a weak or poorly trained model, provide support for the more general view of KD as a form of regularization. Investigation of this general question can be found in others' work as well, *e.g.*, in the context of multi-generational self-KD (Zhang and R. Sabuncu, 2020; Mobahi et al., 2020) and early stopping (Dong et al., 2019).

Finally, a related line of work view KD as "instance-specific label smoothing", where the amount of smoothing is not uniform across categories, but is a function of the input approximated using a pre-trained network (Zhang and R. Sabuncu, 2020; Lee et al., 2022). We note that this is a fundamentally different assertion than KD being a form of LS with a uniform prior, since such methods do in fact bring new task-specific knowledge into the training process, with the pre-trained network essentially serving as a teacher model.

## 3 A Closer Look at the Arguments

As already stated, the focus of this paper is on the specific claim by Yuan et al. (2020) that KD is a form of LS with a uniform prior (Szegedy et al., 2016), a primary argument for which is that their cost functions in Eqs. 1 and 2 have a similar form. A careful examination of the two functions, however, reveals a key difference.

To elaborate, let $H(.)$ represent the entropy of a probability distribution; it then follows that $\forall p \in \Delta^n \backslash \{u\} \; H(u) > H(p)$, where $\Delta^n$ is a probability simplex, $u$ is the uniform distribution and $p$ a

different distribution on $\Delta^n$. Accordingly, minimizing $D_{KL}(u, p)$ in Eq. 1 unconditionally increases $H(p)$, training higher-entropy models than ERM, a fact that we also confirm empirically in §4.

The same is not true for $D_{KL}(p^t, p)$ in Eq. 2, however, since $p^t$ is dependent on the specific selection of the teacher model $t$, and $H(p^t)$ can take on any value between 0 and $H(u)$. Crucially, this means that $H(p^t)$ can be both higher or lower than $H(p)$ if the two respective models were trained separately to minimize $\mathcal{L}_{ERM}$. Unlike in LS, it therefore cannot be guaranteed in KD that minimizing $D_{KL}(p^t, p)$ would increase $H(p)$. In §4, we show how the selection of the teacher model affects the predictive uncertainty of the student, for different combinations of their sizes. Based on the same results, we also comment on the teacher-free KD framework of Yuan et al. (2020).

## 4 Experimental Results

***Setup.*** We use four text classification tasks from GLUE (Wang et al., 2019) in our experiments:

- **MRPC** (Dolan and Brockett, 2005) asks for a *yes/no* decision on if two given sentences are paraphrases of each other.
- **QNLI** provides a question and a sentence from SQuAD (Rajpurkar et al., 2016) and asks if the latter answers the former (*yes/no*).
- **SST-2** (Socher et al., 2013) is a sentiment detection task involving *positive/negative* classification of movie reviews.
- **MNLI** (Williams et al., 2018) is a 3-way (*entailment/contradiction/neutral*) classification task asking for whether a premise sentence entails a hypothesis sentence.

For each task, we use a train-validation-test split for training, model selection, and evaluation, respectively. See Appendix A.1 for more details.

To examine the effect of KD in depth, like Yuan et al. (2020), we run experiments under three different conditions: ***Standard* KD** distills from a fine-tuned BERT-base (Devlin et al., 2019) (BERT henceforth) classifier into a DistilBERT-base (Sanh et al., 2019) (DistilBERT henceforth) student; ***Self*-KD** distills from a DistilBERT teacher into a DistilBERT student; ***Reverse* KD** distills from a DistilBERT teacher into a BERT student. In all cases, the teacher is trained using ERM with a cross-entropy loss.

For every model evaluated under each condition, we perform an exhaustive search over the hyperparameter grid of Table 1 (when applicable).

| Hyperparameter | Values |
| --- | --- |
| Learning rate | $\{1, 2, ..., 7\} \times 10^{-5}$ |
| # of epochs | $\{1, 2, ..., 10\}$ |
| $\alpha$ (Eqs. 1 and 2) | $\{.1, .2, ..., .9\}$ |

Table 1: Hyperparameter grid for model selection.

Batch size is kept fixed at 32. We train with mixed precision on a single NVIDIA A100 GPU. An AdamW optimizer with no warm-up is used. In Appendix A.2, we provide the hyperparameter configuration that is optimal on validation and is subsequently evaluated on test for each model and task. We use the Transformers library of Wolf et al. (2020) for implementation.

The experiments reported in this section use a KD temperature $\tau = 1$, effectively disallowing any artificial smoothening of the target distribution. In Appendix A.3, we also show that treating $\tau$ as a hyperparameter during model selection does not qualitatively change these results.

| Method | MRPC | QNLI | SST-2 | MNLI |
| --- | --- | --- | --- | --- |
| ERM | .1123 | .1419 | .1101 | .2698 |
| LS | .6896 | .6752 | .4447 | .8103 |
| Standard KD | .0548 | .1210 | .0594 | .1887 |
| Self-KD | .0738 | .1341 | .0977 | .2758 |

Table 2: Test set posterior entropies of different DistilBERT classifiers. LS increases entropy over ERM as expected, while KD almost always reduces it.

***Results.*** Table 2 compares the average entropy of posteriors over classes predicted by LS models with those by standard and self-KD students, measured on the four test sets. As expected, LS increases model uncertainty over ERM, in fact quite significantly ($\sim$3–6$\times$). Both KD variants, on the other hand, bring entropy down; the only exception to this is self-KD on MNLI, where we see a very slight increase. These results have direct implications for the central research question of this paper, as they reveal that the typical effect of KD is a reduction in predictive uncertainty over ERM, which is the exact opposite of what LS does *by design*.

Figure 1 illustrates how each training method regulates model uncertainty at real time during training. Both KD variants cause a rapid drop in entropy early in training, and an entropy lower than that of both ERM and LS is maintained throughout. The comparisons are also very similar between the training and the test set, indicating that the smoothing effects (or lack thereof) of each of these methods generalize well from training to inference.

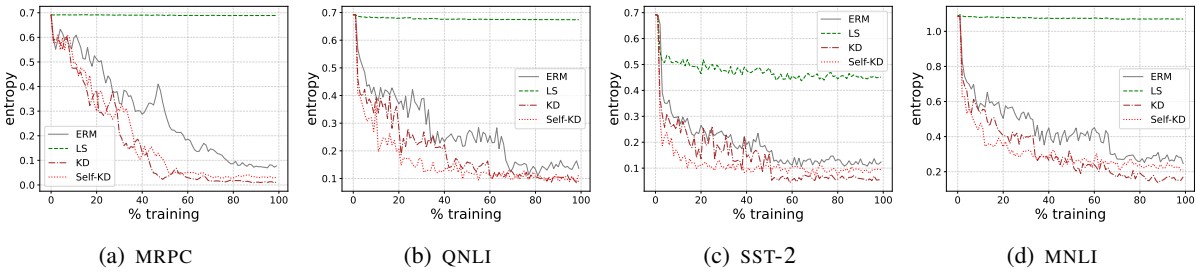

Figure 1: Change in posterior entropy of different DistilBERT classifiers as training progresses.

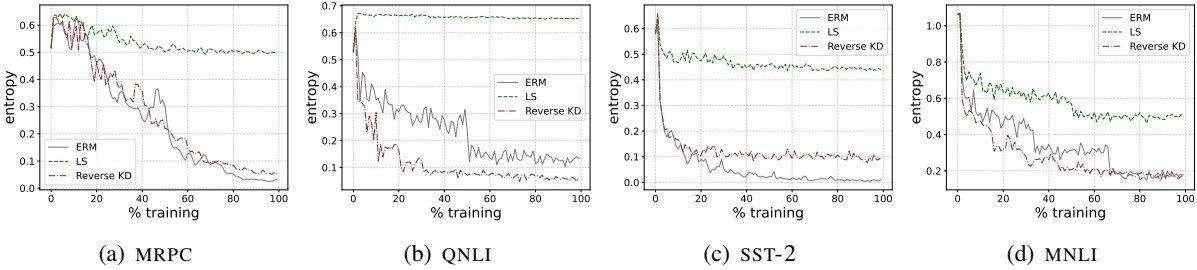

Figure 2: Change in posterior entropy of different BERT classifiers as training progresses.

| Method | MRPC | QNLI | SST-2 | MNLI |
|---|---|---|---|---|
| ERM | 81.0 | 88.5 | 93.8 | 81.3 |
| LS | 80.3 | 88.4 | 95.0 | 81.0 |
| Standard KD | 82.0 | 89.2 | 94.6 | 81.8 |
| Self-KD | 82.3 | 89.0 | 94.7 | 82.2 |

Table 3: Test set accuracy of DistilBERT fine-tuned with ERM, KD and LS. KD has the best performance profile.

| Method | MRPC | QNLI | SST-2 | MNLI |
|---|---|---|---|---|
| ERM | .0678 | .1218 | .0124 | .1845 |
| LS | .5162 | .6528 | .4376 | .4770 |
| Reverse KD | .0954 | .0834 | .0954 | .2041 |

Table 4: Test set posterior entropies of BERT classifiers. KD increases entropy in three out of four tasks, but the LS-induced increases are far greater in magnitude.

In Table 3 we report the test set accuracy of each method on the four tasks. Both KD variants do generally better than LS, as the latter also lags behind the ERM baseline in three out of four evaluations. Interestingly, self-KD outperforms standard KD in most cases. While Yuan et al. (2020) explain similar successes of weaker teacher models in image classification with the equivalence of KD and LS, an alternative explanation is that these results are simply an outcome of the standard-KD student not adequately learning to mimic its higher-capacity teacher due to limited amounts of training data. Others (Liang et al., 2021; Sultan et al., 2022) have shown that such teacher-student knowledge gaps can be reduced by using additional synthetic data during distillation.

Next we turn our attention to reverse KD from DistilBERT into BERT. In Table 4, LS unconditionally increases posterior entropy as before. The effect of reverse KD, on the other hand, is qualitatively different than standard and self-KD, as it also increases entropy in most cases over ERM. We

also observe a similar difference in runtime training entropy, as depicted in Figure 2.

The contrast in the results of Tables 2 and 4 paints a clear and important picture: the predictive uncertainty of the student in KD is a direct function of that of the teacher, and not of the KD process. To elaborate, we first point the reader to row 1 of both tables, which show that when trained using ERM, BERT classifiers have a lower posterior entropy than their smaller DistilBERT counterparts. Consequently, in Table 2, standard KD yields lower-entropy students than self-KD. For the same reason, reverse KD in Table 4 also generally trains higher-entropy models than ERM.

The above results have key implications for the general question concerning the true nature of KD, namely, is it knowledge transfer or regularization? As a student learns to mimic its teacher in KD, we see in the above results that it also inherits its confidence—an integral aspect of the data-driven knowledge of both models. This is clearly unlike methods such as LS or teacher-free KD in (Yuan

| Method | MRPC | QNLI | SST-2 | MNLI |
|--------|------|------|-------|------|
| ERM | 84.0 | 91.0 | 94.5 | 83.7 |
| LS | 84.1 | 89.8 | 94.6 | 83.7 |
| Reverse KD | 83.4 | 91.3 | 94.9 | 83.9 |

Table 5: Test set accuracy of BERT classifiers fine-tuned with different training methods. KD and LS again have generally dissimilar performance profiles.

et al., 2020), which unconditionally reduce confidence in a data-agnostic way to limit overfitting.

With a weaker teacher model, reverse KD yields a smaller overall gain in accuracy than before over ERM and LS (see Table 5). It is interesting, nevertheless, that a lower inductive bias along with strong language modeling capabilities enables the reverse-KD student to outperform its teacher.

## 5 Conclusion

Knowledge distillation is a powerful framework for training lightweight and fast yet high-accuracy models. While still a topic of active interest, any misinterpretation, including false equivalences with other methods, can hinder progress for the study of this promising approach going forward. In this paper, we present evidence against its interpretation as label smoothing, and more generally, regularization. We hope that our work will inspire future efforts to understand distillation better and further improve its utility.

## Limitations

To manage the complexity of our already large-scale experiments involving (a) four different classification tasks, and (b) hyperparameter grids containing up to 1,960 search points, we ran experiments with a single random seed. While this is sufficient for exploring our main research question involving the posterior entropy of different models—for which we only need to show one counterexample, *i.e.*, one where entropy decreases with knowledge distillation—the exact accuracy figures would be more reliable if averaged over multiple seeds.

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

# A   Appendix

## A.1   Datasets Used in Our Experiments

We collect all datasets and their splits from the Hugging Face Datasets library[1]. Among the four tasks, only MRPC has a labeled test set. For the other three datasets, we split the train set into two parts. In QNLI and MNLI experiments, the two parts are used for training and validation, and the original validation set is used as a blind test set. For SST-2, since the validation set is small, we also use it for model selection; the train set is therefore split into train and test sets. For MNLI, we evaluate only on the `test-matched` set, which consists of unseen in-domain examples. Table 6 shows the number of examples in each dataset.

| Task | Train | Validation | Test |
|------|-------|------------|------|
| MRPC | 3,668 | 408 | 1,725 |
| QNLI | 99,743 | 5,000 | 5,463 |
| SST-2 | 65,549 | 872 | 1,800 |
| MNLI | 382,887 | 9,815 | 9,815 |

Table 6: Size statistics for our datasets.

## A.2   Model Selection

Table 7 contains the optimal configurations for the models evaluated in §4 as observed on the respective validations sets.

## A.3   Distillation at Higher Temperatures

We present results for the same experiments as in §4 here, with two main differences: First, for KD, we include the temperature $\tau$ in the hyperparameter search and re-define the domain for $\alpha$ as follows:

| Hyperparameter | Values |
|----------------|--------|
| $\tau$ | $\{1, 2, 4, 8, 16, 32, 64\}$ |
| $\alpha$ (Eq. 2) | $\{.25, .5, .75, 1\}$ |
| Learning rate | $\{1, 2, ..., 7\} \times 10^{-5}$ |
| # of epochs | $\{1, 2, ..., 10\}$ |

yielding a 4D grid consisting of 1,960 points. Note that for this more general experiment, we allow $\alpha = 1$, which only uses the teacher's soft targets for training. Second, to keep the grid size the same (1,960) for LS, we re-define the domain of $\alpha$ for LS to contain 28 equally-distanced points ranging from .1 to .9991.

In Table 8 and Figure 3, we report posterior entropies of DistilBERT models measured at inference and during training, respectively. Similar to the experimental results in §4, both KD variants generally train models with lower entropy than not

only LS, but also ERM. In Table 9, KD again has better overall test set accuracy than LS. Across all these results, the comparisons between standard and self-KD also remain generally unchanged.

The reverse KD results—for both predictive uncertainty in Table 10 and Figure 4, and test set performance in Table 11—also show very similar trends as those described in §4. Altogether, the results presented in this section with KD in three different settings demonstrate that even at high temperatures, our original findings and conclusions hold true.

The optimal hyperparameter combinations for the above experiments are provided in Table 12.

---

[1] https://huggingface.co/docs/datasets/index

| Task | Model | Training | $\alpha$ | Learning Rate | # Epochs |
|------|-------|----------|----------|---------------|----------|
| MRPC | BERT | ERM | — | 7e-5 | 4 |
| | | LS | .4 | 7e-5 | 7 |
| | | Reverse KD | .1 | 4e-5 | 5 |
| | DistilBERT | ERM | — | 5e-5 | 4 |
| | | LS | .9 | 4e-5 | 4 |
| | | Standard KD | .1 | 6e-5 | 7 |
| | | Self-KD | .8 | 7e-5 | 6 |
| QNLI | BERT | ERM | — | 4e-5 | 2 |
| | | LS | .7 | 2e-5 | 3 |
| | | Reverse KD | .3 | 2e-5 | 10 |
| | DistilBERT | ERM | — | 3e-5 | 3 |
| | | LS | .8 | 2e-5 | 8 |
| | | Standard KD | .8 | 3e-5 | 5 |
| | | Self-KD | .7 | 2e-5 | 10 |
| SST-2 | BERT | ERM | — | 1e-5 | 9 |
| | | LS | .3 | 2e-5 | 3 |
| | | Reverse KD | .7 | 1e-5 | 7 |
| | DistilBERT | ERM | — | 1e-5 | 2 |
| | | LS | .3 | 6e-5 | 2 |
| | | Standard KD | .6 | 6e-5 | 2 |
| | | Self-KD | .7 | 5e-5 | 8 |
| MNLI | BERT | ERM | — | 3e-5 | 3 |
| | | LS | .1 | 3e-5 | 2 |
| | | Reverse KD | .4 | 3e-5 | 7 |
| | DistilBERT | ERM | — | 3e-5 | 3 |
| | | LS | .8 | 3e-5 | 3 |
| | | Standard KD | .7 | 3e-5 | 6 |
| | | Self-KD | .9 | 3e-5 | 9 |

Table 7: Optimal hyperparameter combinations ($\tau$=1) for the §4 experiments.

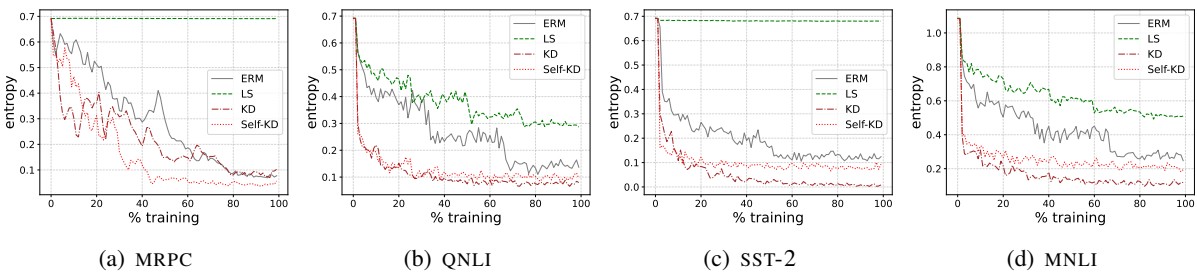

| (a) MRPC | (b) QNLI | (c) SST-2 | (d) MNLI |
|----------|----------|-----------|----------|

Figure 3: Change in posterior entropy for different DistilBERT classifiers as training (at optimal $\tau$) progresses.

| Method | MRPC | QNLI | SST-2 | MNLI |
|--------|------|------|-------|------|
| ERM | .1123 | .1419 | .1101 | .2698 |
| LS | .6916 | .2949 | .6802 | .5095 |
| Standard KD | .1067 | .1024 | .0107 | .1481 |
| Self-KD | .0927 | .1179 | .0853 | .2476 |

Table 8: Test set posterior entropies of DistilBERT classifiers fine-tuned with different training methods. Label smoothing (LS) increases entropy over ERM as expected, but distillation (KD) induces a drop on all tasks.

| Method | MRPC | QNLI | SST-2 | MNLI |
|--------|------|------|-------|------|
| ERM | .0678 | .1218 | .0124 | .1844 |
| LS | .3778 | .6512 | .4891 | .4770 |
| Reverse KD | .1058 | .1140 | .0995 | .2163 |

Table 10: Test set posterior entropies of BERT classifiers fine-tuned with different training methods. While KD does increase entropy over ERM on three out of four tasks, LS yields increases that are much higher.

| Method | MRPC | QNLI | SST-2 | MNLI |
|--------|------|------|-------|------|
| ERM | 81.0 | 88.5 | 93.8 | 81.3 |
| LS | 78.4 | 88.8 | 95.2 | 80.8 |
| Standard KD | 80.2 | 89.6 | 94.6 | 82.7 |
| Self-KD | 82.1 | 89.5 | 94.8 | 82.6 |

Table 9: Test set accuracy of DistilBERT classifiers fine-tuned with different training methods. KD has the best overall performance profile.

| Method | MRPC | QNLI | SST-2 | MNLI |
|--------|------|------|-------|------|
| ERM | 84.0 | 91.0 | 94.5 | 83.7 |
| LS | 83.9 | 90.8 | 94.5 | 83.7 |
| Reverse KD | 84.3 | 91.3 | 94.9 | 84.0 |

Table 11: Test set accuracy of BERT classifiers fine-tuned with different training methods. KD clearly outperforms LS.

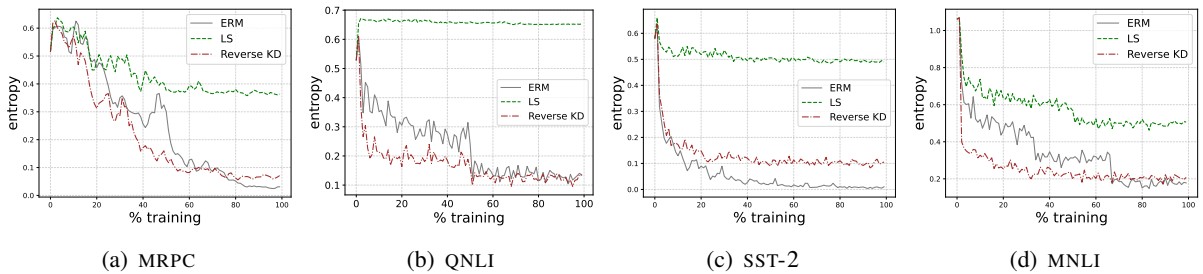

(a) MRPC    (b) QNLI    (c) SST-2    (d) MNLI

Figure 4: Change in posterior entropy for different BERT classifiers as training (at optimal $\tau$) progresses.

| Task | Model | Training | $\alpha$ | $\tau$ | Learning Rate | # Epochs |
|---|---|---|---|---|---|---|
| MRPC | BERT | ERM | — | — | 7e-5 | 4 |
| | | LS | .2332 | — | 7e-5 | 6 |
| | | Reverse KD | 1 | 1 | 4e-5 | 6 |
| | DistilBERT | ERM | — | — | 5e-5 | 4 |
| | | LS | .9325 | — | 3e-5 | 5 |
| | | Standard KD | 1 | 16 | 3e-5 | 3 |
| | | Self-KD | .25 | 2 | 7e-5 | 7 |
| QNLI | BERT | ERM | — | — | 4e-5 | 2 |
| | | LS | .6994 | — | 4e-5 | 3 |
| | | Reverse KD | .25 | 64 | 4e-5 | 2 |
| | DistilBERT | ERM | — | — | 3e-5 | 3 |
| | | LS | .1333 | — | 3e-5 | 4 |
| | | Standard KD | .75 | 64 | 3e-5 | 9 |
| | | Self-KD | .5 | 64 | 3e-5 | 8 |
| SST-2 | BERT | ERM | — | — | 1e-5 | 9 |
| | | LS | .3664 | — | 2e-5 | 3 |
| | | Reverse KD | .75 | 1 | 1e-5 | 5 |
| | DistilBERT | ERM | — | — | 1e-5 | 2 |
| | | LS | .8326 | — | 5e-5 | 3 |
| | | Standard KD | .5 | 1 | 2e-5 | 9 |
| | | Self-KD | .25 | 8 | 5e-5 | 6 |
| MNLI | BERT | ERM | — | — | 3e-5 | 3 |
| | | LS | .1 | — | 3e-5 | 2 |
| | | Reverse KD | .5 | 4 | 2e-5 | 7 |
| | DistilBERT | ERM | — | — | 3e-5 | 3 |
| | | LS | .1666 | — | 3e-5 | 5 |
| | | Standard KD | 1 | 4 | 4e-5 | 10 |
| | | Self-KD | .75 | 64 | 5e-5 | 9 |

Table 12: Optimal hyperparameter combinations for the §A.3 experiments.