# OpenReview forum: "Knowledge Distillation ≈ Label Smoothing: Fact or Fallacy?"
_EMNLP/2023/Conference — EMNLP 2023 Main_

### Official Review · Reviewer_FFtC · 2023-07-31

**Soundness:** 4

**Excitement:**

4: Strong: This paper deepens the understanding of some phenomenon or lowers the barriers to an existing research direction.

**Paper Topic And Main Contributions:**

This paper studies the relationshiop between Knowledge Distillation (KD) and Label Smoothing (LS). The authors re-examine the claim that "KD is a type of learned LS regularization" and claim that KD and LS work in different ways and are therefore not equivalent. They empirically show that KD and LS are only similar in terms of the loss function form. Experiments demonstrate that LS largely increases the model uncertainty over standard empirical risk minimization while KD tends to decrease it. This paper provides a different view of KD vs. LS, which may lead to better understanding of KD in future research.

**Reasons To Accept:**

- This paper gives a different point of view against the previous studies that "KD is a form of LS regularization". The authors give analyses that KD and LS are different, and provide empirical results to support their arguments.
- The reverse KD experiments is also interesting, and show that KD is more about transferring knowledge from a teacher model to a student, rather than a regularization process like LS.
- This paper may give others a better understanding of KD in future research.

**Reasons To Reject:**

- The experimental tasks in this paper is multiple text classification tasks, while the task in Yuan et al., 2020 is image classification. I was wondering whether the different conclusions drawn from this paper and Yuan et al., 2020 are due to the totally different tasks.

**Reproducibility:**

4: Could mostly reproduce the results, but there may be some variation because of sample variance or minor variations in their interpretation of the protocol or method.

**Reviewer Confidence:**

3: Pretty sure, but there's a chance I missed something. Although I have a good feel for this area in general, I did not carefully check the paper's details, e.g., the math, experimental design, or novelty.

**Typos Grammar Style And Presentation Improvements:**

- L41: Maybe an explanation for NLP before using the abbreviation.

---

> ### Author Rebuttal · Authors · 2023-08-29
>
> We thank Reviewer FFtC for their valuable feedback. We respond to their questions and concerns below.
>
> **RR1:**  The experimental tasks in this paper is multiple text classification tasks, while the task in Yuan et al., 2020 is image classification. I was wondering whether the different conclusions drawn from this paper and Yuan et al., 2020 are due to the totally different tasks.
>
> **Response:** An inductive claim such as KD ≈ LS must hold across different tasks and modalities. The fact that in text classification we observe very different characteristics in the models trained by the two methods already sufficiently refutes the equivalence assertion. That said, it is also not difficult to run a thought experiment on image classification where we can expect to see similar results: First, LS by design would always train high-entropy models irrespective of task and modality. All that is now needed to complete this thought experiment is an overconfident image classification teacher model (already quite common among large neural networks, but can also be created artificially using techniques such as temperature downscaling) for KD -- distilling such a teacher at low temperatures, when successful, should yield students that are also peaky. Thus we strongly believe that the different conclusions reached by Yuan et al. and us can be attributed to the different levels of detail in which the two studies explore the KD ≈ LS claim and are not due to our divergent selection of tasks.

---

### Official Review · Reviewer_qYKw · 2023-08-04

**Soundness:** 4

**Excitement:**

3: Ambivalent: It has merits (e.g., it reports state-of-the-art results, the idea is nice), but there are key weaknesses (e.g., it describes incremental work), and it can significantly benefit from another round of revision. However, I won't object to accepting it if my co-reviewers champion it.

**Paper Topic And Main Contributions:**

This paper is about the interpretation of knowledge distillation (KD) in machine learning and its relationship to label smoothing (LS). The paper addresses the problem of misinterpretation of KD as a form of regularization, and the false equivalences made between KD and LS. The main contribution of the paper is to provide evidence against the interpretation of KD as LS, and to clarify the fundamental differences between the two techniques.

**Reasons To Accept:**

The paper is clear and well-written. This paper shows an investigation of the relationship between knowledge distillation and label smoothing, which can help the community to better understand the strength of knowledge distillation. The experimental results provide valuable insights into the behavior of these techniques in text classification tasks and contribute to a better understanding of knowledge distillation.


**Reasons To Reject:**

1. The paper will be stronger if the analysis is extended to other machine learning tasks such as image classification in CV.
2. Though this paper shows the advantage of KD over LS, the results have been known by most researchers about KD (since they can simply get the conclusions by the model performance). I wonder whether there are any suggestions for readers to design a better knowledge distillation model in application.

**Reproducibility:**

4: Could mostly reproduce the results, but there may be some variation because of sample variance or minor variations in their interpretation of the protocol or method.

**Reviewer Confidence:**

4: Quite sure. I tried to check the important points carefully. It's unlikely, though conceivable, that I missed something that should affect my ratings.

---

> ### Author Rebuttal · Authors · 2023-08-29
>
> We thank Reviewer qYKw for their valuable feedback. We respond to their questions and concerns below.
>
> **RR1:** "The paper will be stronger if the analysis is extended to other machine learning tasks such as image classification in CV."
>
> **Response:** While we agree that results on more tasks and modalities would be interesting to look at, all that is needed to refute an inductive claim such as KD ≈ LS is a single counterexample. The goal of this short paper submission to EMNLP is to present a counter-perspective to the above claim in the modality of language, which we do using experiments on four different text classification tasks. We hope that our work will inspire future efforts to understand KD better, including on other tasks and modalities. Please also see our response to Reviewer FFtC for a related discussion.
>
> **RR2:** "Though this paper shows the advantage of KD over LS, the results have been known by most researchers about KD (since they can simply get the conclusions by the model performance). I wonder whether there are any suggestions for readers to design a better knowledge distillation model in application."
>
> **Response:** We would like to first clarify that our primary goal is not to show that KD is a superior model supervision technique to LS, but that their underlying mechanisms are different. This is part of the more general question of correctly interpreting KD, the importance of which we discuss in Sections 1 and 5. Given this goal, developing better KD algorithms is out of the scope of this short paper submission.

---

### Official Review · Reviewer_39Vd · 2023-08-06

**Soundness:** 2

**Excitement:**

3: Ambivalent: It has merits (e.g., it reports state-of-the-art results, the idea is nice), but there are key weaknesses (e.g., it describes incremental work), and it can significantly benefit from another round of revision. However, I won't object to accepting it if my co-reviewers champion it.

**Missing References:**

Lee, Dongkyu, Ka Chun Cheung, and Nevin Zhang. "Adaptive Label Smoothing with Self-Knowledge in Natural Language Generation." Proceedings of the 2022 Conference on Empirical Methods in Natural Language Processing. 2022.

Yun, Sukmin, et al. "Regularizing class-wise predictions via self-knowledge distillation." Proceedings of the IEEE/CVF conference on computer vision and pattern recognition. 2020.

**Paper Topic And Main Contributions:**

The paper revisits the connection between knowledge distillation (KD) and label smoothing (LS). Previous studies suggested that KD could be interpreted as a form of LS. Both KD and LS introduce soft labels instead one-hot targets. By minimizing the KL divergence between the student's predictions and soft labels, KD acts as a regularizer similar to how LS does with its smoothed labels. However, in this study, the authors argue that KD and LS are not equivalence. The authors evaluate this through four text classification tasks from the GLUE benchmark. By analyzing the entropies of LS and KD, the experimental results present evidence against KD's interpretation as label smoothing.

**Questions For The Authors:**

Question A: Line 151. Why does minimizing D_KL(u, p) in Eq. 1 increases H(p)?
As D_KL(u,p) = H(u,p) - H(u), there is no guarantee that minimizing D_KL(u,p) necessarily results in an increase of H(p).

**Reasons To Accept:**

- The experimental findings, which suggest that the student inherits the teacher's predictive uncertainty, are somewhat interesting.

- The experiment is well organized, showing the changes in entropy and accuracy throughout training and testing.

**Reasons To Reject:**

- The claim is grounded in empirical findings and does not provide a solid mathematical foundation.

-  Although I acknowledge that KD and LS are not identical, I believe KD can be viewed as a special form of LS. This is particularly true when the teacher network is uniformly distributed and the temperature is set at 1, then LS and KD are equivalent.

- The authors only compared one of the existing works in this area and did not sufficiently address related works.
Here are some related works for LS and KD:

Lee, Dongkyu, Ka Chun Cheung, and Nevin Zhang. "Adaptive Label Smoothing with Self-Knowledge in Natural Language Generation." Proceedings of the 2022 Conference on Empirical Methods in Natural Language Processing. 2022.

Zhang, Zhilu, and Mert Sabuncu. "Self-distillation as instance-specific label smoothing." Advances in Neural Information Processing Systems 33 (2020): 2184-2195.

Li Yuan, Francis EH Tay, Guilin Li, Tao Wang, and Jiashi Feng. "Revisit knowledge distillation: a teacher-free framework." arXiv preprint arXiv:1909.11723, 2019.

Yun, Sukmin, et al. "Regularizing class-wise predictions via self-knowledge distillation." Proceedings of the IEEE/CVF conference on computer vision and pattern recognition. 2020.


**Reproducibility:**

4: Could mostly reproduce the results, but there may be some variation because of sample variance or minor variations in their interpretation of the protocol or method.

**Reviewer Confidence:**

5: Positive that my evaluation is correct. I read the paper very carefully and I am very familiar with related work.

---

> ### Author Rebuttal · Authors · 2023-08-29
>
> We thank Reviewer 39Vd for their valuable feedback. We respond to their questions and concerns below.
>
> **RR1:** "The claim is grounded in empirical findings and does not provide a solid mathematical foundation."
>
> **Response:** We use a combination of mathematical arguments (Section 3) and empirical analysis (Section 4) to refute the existing claim that knowledge distillation (KD) is a form of label smoothing (LS) regularization. Reviewer 39Vd is correct to point out that our argument has empirical components, but it is unclear why that makes a scientifically flawed approach, as proof by counterexamples is a well-established method used to refute inductive claims such as KD ≈ LS.
>
> **RR2:** "... KD can be viewed as a special form of LS. This is particularly true when the teacher network is uniformly distributed and the temperature is set at 1, then LS and KD are equivalent."
>
> **Response:** Since a strictly uniform teacher does not provide any useful training signal, we assume that Reviewer 39Vd meant a smoothened teacher. Since Reviewer 39Vd also referred to a "teacher network", we assume that LS is being applied to smoothen the output of a teacher network previously trained on labeled data. Such a teacher would combine two separate notions: (a) sourcing the target distribution from a trained network as in KD, and (b) smoothing that distribution with LS. We must recognize that this is a combination of KD and LS, not LS playing the role of KD.
>
> **RR3:** "The authors only compared one of the existing works in this area and did not sufficiently address related works. Here are some related works for LS and KD: ..."
>
> **Response:** One of the papers Reviewer 39Vd lists (“Revisit Knowledge Distillation: a Teacher-free framework”) is actually an earlier version of our baseline paper by Yuan et al. For the rest, it is unclear to us what exactly to compare with from these papers. We have a very specific goal: to examine the accuracy of the assertion that KD and LS are *equivalent*, and not any other relation between the two methods or how such relations can be exploited to develop new algorithms such as "teacher-free distillation". While these papers generally explore KD and LS/regularization in some way, none of them directly investigate the question of whether LS and KD are equivalent. Crucially, these papers also do not examine the properties of what is arguably the most common form of KD — distilling a bigger teacher model into a smaller student model — where we find the differences between KD and LS to be the starkest. We are generally aware of these studies, and will cite them and more as related work in the final version given the additional space.
>
> **Q(A):** "... Why does minimizing D_KL(u, p) in Eq. 1 increases H(p)? As D_KL(u,p) = H(u,p) - H(u), there is no guarantee that minimizing D_KL(u,p) necessarily results in an increase of H(p)."
>
> **Response:** The uniform distribution u has the highest entropy H(u) among all distributions on the respective probability simplex. Accordingly, as p becomes smoother and H(p) increases, H(u, p) goes down, also lowering D_KL(u, p) = H(u, p) - H(u) [Note that H(u) is a constant]. The following table illustrates this for the Bernoulli distribution (the distribution associated with binary classification tasks) as an example:
>
> | p(0)  | p(1) | H(p) | H(u, p) | D_KL(u, p) |
> | :-: | :-: |:-: | :-: | :-: |
> | .1 | .9 | .141 | .523 | .222 |
> | .2 | .8 | .217 | .398 | .097 |
> | .3 | .7 | .265 | .339 | .038 |
> | .4 | .6 | .292 | .310 | .009 |
> | .5 | .5 | .301 | .301 | 0 |
> | .6 | .4 | .292 | .310 | .009 |
> | .7 | .3 | .265 | .339 | .038 |
> | .8 | .2 | .217 | .398 | .097 |
> | .9 | .1 | .141 | .523 | .222 |

---

### Meta-Review · Area_Chair_YYMo · 2023-09-05

**Recommendation:** 5
**Best Paper Recommendation:** No

**Metareview:**

The paper provides empirical evidence that knowledge distillation and label smoothing are not equivalent, contradicting previous work which has argued that KD could be interpreted as LS. The reviewers are generally in agreement that the empirical results are interesting and convincing, and that the experimental setup is well-executed. However R1 believes that the paper could benefit from a theoretical treatment to support the empirical evidence in order to be more convincing, as well as addressing more of the literature on the topic. Additionally, the generality of the results could be improved by incorporating data with different modalities e.g. vision. Overall though, the reception is quite positive and the paper would make a good contribution to the conference.

**Meta-Review:**

The paper provides empirical evidence that knowledge distillation and label smoothing are not equivalent, contradicting previous work which has argued that KD could be interpreted as LS. The reviewers are generally in agreement that the empirical results are interesting and convincing, and that the experimental setup is well-executed. However R1 believes that the paper could benefit from a theoretical treatment to support the empirical evidence in order to be more convincing, as well as addressing more of the literature on the topic. Additionally, the generality of the results could be improved by incorporating data with different modalities e.g. vision. Overall though, the reception is quite positive and the paper would make a good contribution to the conference.

---

### Decision · Program_Chairs · 2023-10-07

**Decision:**

Accept-Main

**Comment:**

The paper provides empirical evidence that knowledge distillation and label smoothing are not equivalent, contradicting previous work which has argued that KD could be interpreted as LS. The reviewers are generally in agreement that the empirical results are interesting and convincing, and that the experimental setup is well-executed. However R1 believes that the paper could benefit from a theoretical treatment to support the empirical evidence in order to be more convincing, as well as addressing more of the literature on the topic. Additionally, the generality of the results could be improved by incorporating data with different modalities e.g. vision. Overall though, the reception is quite positive and the paper would make a good contribution to the conference.